# Time-motion study in primary health care in moldova: How do family doctors and medical assistants spend their work time?

Altiona Muho[1,2]*, Jari Kempers[3], Alexandra Topa[4], Ala Curteanu[5,6], Ghenadie Curocichin[4], Helen Prytherch[1,2]

1 Swiss Centre for International Health, Swiss Tropical and Public Health Institute, Allschwil, Switzerland, 2 University of Basel, Basel, Switzerland, 3 European Health Economics Oy, Helsinki, Finland, 4 Family Medicine Department, State University of Medicine and Pharmacy "Nicolae Testemitanu", Chisinau, Republic of Moldova, 5 Healthy Life Project, Chisinau: Reducing the Burden of Noncommunicable Diseases, Chisinau, Republic of Moldova, 6 Research and Innovation Department, Mother and Child Health Institute, Chisinau, Republic of Moldova

* altiona.muho@swisstph.ch

## Abstract

### Introduction

Despite efforts to strengthen primary health care, Moldova faces continues challenges related to health personnel shortages and distribution disparities, resulting in high workloads. To better understand the current situation, we sought to assess the work-time allocation of family doctors and medical assistants in primary health care centers in Moldova.

### Methods

An observational, cross-sectional study was conducted to investigate health personnel's time allocated to activity categories, the number and duration of consultations for different health conditions, and the reasons for patient's visits in nine primary health centers in Moldova. 24 family doctors and 24 medical assistants were observed for five consecutive working days, covering in total 233 workdays and 4,916 consultations.

### Results

Family doctors used 49.2% of their time on direct patient care, 20.4% on administrative tasks and 9.8% on outreach activities. Medical assistants spent 33.6% on administration tasks, 30.9% on direct patient care and 13.6% on outreach activities. Of their direct patient care time, family doctors and medical assistants used 49.8% and 51.7%, respectively, on non-communicable disease consultations. The median duration of doctors' consultations was 8.55 minutes, whereas medical assistants' consultations lasted 5.65 minutes. Hypertensive disease and pediatric visits were the most frequent and time-consuming consultations. Health personnel contracted for 1.5 full-time equivalents worked over 10 hours per week shorter than required by their working contracts.

**Data availability statement:** Requirement acknowledged. We provide hereby an institutional point of contact and cite where the data can be found. Data cannot be shared publicly as it contains information that could compromise the privacy of research participants. Additionally, consent for publication of the data was not obtained as it posed the risk of potential participants to decline joining the study, due to the sensitive nature of the data gathered. Furthermore, due to the small size of the HCs where the study was conducted, there is a slight possibility of the results being traced back to the participants. However, data can become available upon reasonable request from our institutional point of contact open-science@swisstph.ch. Furthermore, the data is stored on a secure server in the Swiss TPH datacenter located in Allschwil, Switzerland.

**Funding:** The Swiss Agency for Development (SDC) provided financial support to this implementation study in the context of the Healthy Life Project to reduce the burden of Noncommunicable Diseases (7F-06103.02), Phase II, 2020-2024. The funders had no role in study design, data collection and analysis, decision to publish, or preparation of the manuscript.

**Competing interests:** The authors declare that they have no competing interests.

## Conclusions

Family doctors used nearly half of their work time on direct patient care, while medical assistants spent less than one third, with administrative tasks taking up most of medical assistants' time. Consultation durations were short for both cadres, especially among medical assistants, reflecting the high workload in primary health care in Moldova. Additionally, health personnel contracted for 1.5 full-time equivalents worked consistently fewer hours than required by their contracts, indicating this system does not improve patient access or resolve personnel shortages.

## Introduction

Human resources for health are facing significant challenges, which are particularly pronounced in low and middle-income countries (LMIC), like Moldova. These challenges include workforce shortages, widespread migration of health professionals, geographical imbalances, skill mix disparities, and inefficient management practices [1–4]. Collectively, these challenges contribute to a strain on the workforce performance, especially at Primary Health Care (PHC) level, and negatively affect the efficiency of health service delivery. The requirements of increasingly complex patient care further aggravate this situation in particular where it is difficult for communities to access higher levels of care [5]. Addressing these issues is essential, as highlighted in the Global Human Resources for Health Strategy [2].

Productivity is a key driver of workforce performance [6] and as such, requires comprehensive measures for evaluation. One particular measure is health providers' use of time, which needs to be compared against benchmarks that derive from duties defined in job descriptions [7]. Time motion studies are the primary method for appraising how health providers allocate their work time.

Time motion studies are used to analyze working patterns by dividing work into categories of activities and measuring the time that each activity takes up [8]. One of the most accurate technique to conduct time motion studies is continuous observation [9,10]. During the continuous observation, the observer shadows the observed individual, keeps track of tasks that are being performed and records the time that each task takes. While the evidence on health providers' time in health services is increasing worldwide, there are only a few studies in the Eastern European region [8]. This study provides new evidence on the time use of health personnel (HP) in Moldova.

Moldova has experienced three decades of continuous health system reforms since declaring its independence [11]. Over this period, there has been a notable emphasis on enhancing the PHC system, manifested through decentralization, increased funding allocations, and the assignment of a gate-keeping role to PHC, with family medicine positioned at its core [12]. Currently, the PHC services in Moldova are delivered through a network of urban and rural health centers, offices of family doctors (FDs) and health offices and are governed by the Ministry of Health of Republic of Moldova and the National Health Insurance Company [13].

Despite the increased focus on PHC, challenges persist in the form of HP shortages, most notably of FDs, due also to the lack of professional recognition of this role, and their regional distribution disparities, which pose human resource management issues within the health system. Much of Moldova is rural which is where the shortages make themselves most felt, while it is these communities that have least recourse to other levels of care, or can only try to do so by investing their own time and money to travel to urban centers. In 2022, nearly a thousand FD positions and over a thousand MA positions were unfilled. This followed the period from 2018 to

2021, during which the health system lost over 1,200 FDs and MAs, reducing the number of FDs in the country by 6.6% [14]. The majority of FDs cover a catchment population of 3,000–6,000, far exceeding the recommended 1,500 persons per FD [15]. Being overburden has significantly contributed to FDs' job dissatisfaction levels, a concern that has been reportedly documented for the past 10 years in Moldova [16,17]. While Moldova's PHC system faces understaffing and under-resourcing, the available resources are also underutilized. These issues necessitate a detailed examination of health workers' productivity, especially their work-time utilization.

At present, little is known about work-time allocation of FDs and MAs in PHC centers of Moldova. This study seeks to answer the research question: How do FDs and MAs spend their work time in PHC centers in Moldova? It investigates the allocation of work hours across various activities, patients' health conditions and reasons for visits, and the duration of these consultations. This time motion study is essential for understanding how PHC personnel utilizes the time, highlighting inefficiencies, workload distributions and the time required for various clinical and administrative tasks. These findings can inform targeted interventions by policymakers and healthcare managers to optimize time use, reduce administrative burdens and enhance quality of care. Moreover this study contributes to the evidence base needed to address workforce shortages and improve overall PHC efficiency in Moldova.

## Methods

### Selection of primary health care centers

The time-motion study was carried out in nine outpatient PHC centers. Four centers were chosen based on their participation in an earlier time-use survey in 2018 [18]. The remaining five were selected based on similar criteria: 1) location (district center or rural), 2) catchment populations, and 3) numbers of FDs and MAs. Table 1 provides an overview of PHC centers.

### Healthcare personnel

The participation of HP in the study was voluntary. To increase their willingness to participate, a letter was sent from the Ministry of Health to the managers of PHCs, informing them about the importance of the study and requesting their cooperation. Upon arrival at the HCs, the observers informed the HP about the objectives and methodology of the study. The personnel

**Table 1. Overview of the primary health care centers, healthcare personnel, and consultations observed in the time-motion study.**

| Primary health care centers | Location | Catchment population | Family doctors | Medical assistants | Family doctor consultations | Medical assistant consultations | Total consulta-tions |
|---|---|---|---|---|---|---|---|
| Briceni | District center | 9,598 | 5 | 5 | 540 | 411 | 951 |
| Criuleni | District center | 7,919 | 4 | 4 | 472 | 445 | 917 |
| Mărăndeni, Fălesti | Rural | 2,672 | 1 | 1 | 116 | 93 | 209 |
| Ciolacul Nou, Fălești | Rural | 1,051 | 1 | 1 | 56 | 66 | 122 |
| Rîșcani | District center | 11,142 | 2 | 2 | 212 | 196 | 408 |
| Dondușeni | District center | 8,514 | 3 | 3 | 409 | 271 | 680 |
| Nisporeni | District center | 11,903 | 5 | 5 | 529 | 430 | 959 |
| Corlăteni, Rîșcani | Rural | 5,476 | 2 | 2 | 236 | 248 | 484 |
| Recea, Rîșcani | Rural | 2,551 | 1 | 1 | 107 | 79 | 186 |
| **Total** | | **60,826** | **24** | **24** | **2,677** | **2,239** | **4,916** |

Note: This data reflects the time period 3rd October– 4th November 2022.

were informed that their participation was voluntary and that no individual or healthcare level analyses would be conducted or reported. The observers obtained verbal consent from the HP prior to the start of the study. Following consent, the process began with a form gathering HP general information, which also served as documentation of their voluntary participation. For those who disagreed, no information was collected. Further, HP were recruited upon their availability on the first day the study team visited each PHC center to conduct the observations. This corresponded with every Monday of October 2022, five in total, from 3rd of October until 4th of November. While multiple HP is involved in patient examination process at PHC centers, our study specifically focused on FDs and MAs and we did not collect data on other HP.

In total, the research involved 24 FDs and 24 MAs, who were observed during 4,916 consultations. The number of FDs, MAs, and observed consultations per PHC center are shown in Table 1. FDs are trained physicians with medical degrees and specialized training in family medicine, whereas MAs in Moldova typically complete vocational or post-secondary education programs focused on PHC. MAs in Moldova are comparable to PHC nurses in other health systems. Their role includes supporting FDs by carrying out prophylactic, diagnostic and curative assessments, managing patient records and preparing the workplace and instruments. MAs and FDs typically collaborate in sequential workflow. MAs handle the above-mentioned preparatory tasks with patients prior to FD's consultation, allowing FDs to focus on diagnosis, treatment decisions and complex case management during the interactions with patients.

In our study, most FDs and MAs had extensive experience in medicine, with FDs typically having a longer experience in family medicine. The majority of the HP functioned as regular FDs or MAs supporting FDs, with three holding management roles. Both groups self-reported similar weekly working hours (the observed working hours are detailed in the results section). In the observed PHCs in Moldova, FDs and MAs primarily operated within fixed office hours rather than shift work. Their schedules typically followed a standard workday structure with occasional variations for specific tasks or patient demands. Table 2 provides a summary of the characteristics of the HP. The privacy of health personnel was maintained by excluding their names from the reports and by refraining from conducting individual or health center-level analyses.

## Patient consent and ethical clearance

Patient consent was obtained before consultations. First, patients received a verbal explanation about the research's purpose and were assured that no personal data would be collected, except for age and gender. This ensured that observation data remained untraceable to

**Table 2. Experience, roles, and self-reported working hours of observed healthcare professionals.**

| Characteristics | Family doctors | Medical assistants |
|---|---|---|
| Experience in medicine, years* | 34.08 (11.02) | 31.42 (10.09) |
| Experience in family medicine, years* | 31.67 (11.25) | 22.25 (8.14) |
| Role in primary health care centers* | | |
| Manager | 2 | NA |
| FD | 23 | NA |
| Head MA | NA | 1 |
| MA of a FD | NA | 23 |
| Triage MA | NA | 1 |
| Self-reported working hours per week** | 40.62 (7.94) | 40.38 (8.11) |

*Multiple answers possible.

**Mean (standard deviation).

individual patients. If a patient declined, no observations were conducted inside the consultation room. In these cases, only the duration of the consultation was recorded.

Ethical clearance for this study was granted by the Moldovan Ministry of Health on 25th August 2021 (Decision number 117). The Ministry of Health issued an approval letter to the managers of the PHC facilities, explaining the study's objectives and significance while seeking their cooperation.

## Data collection tools

The study utilized purpose-built electronic forms developed and tested specifically for this study: 1) a form for general information about health personnel, 2) a family doctor observation form, and 3) a medical assistant observation form. These tools underwent a two-stage validation process: initially, over a two-week period, three FDs and three MAs were observed during real consultations at two primary health care centers in Chisinau; subsequently, academic collaborators from the Nicolae Testemiţanu State University of Medicine and Pharmacy tested the forms in four simulated consultations. The forms were in Romanian and filled out solely by the observers using tablets. HP or patients were not permitted to use the forms.

Prior to the commencement of the observations, the general information form was used to collect details such as age, medical education, and role at the PHC center. Data on self-reported working hours and employment contract types were also collected. This information was used to enrich the analyses.

The observation forms, for both FDs and MAs, follow a similar structure. Each form begins by recording the start time and the current task engaged in by the FD or MA. Depending on the type of activity, specific follow-up fields are presented. These fields are then filled out, and finally, the end time of the activity is recorded.

Further, the observation time can be distinguished as consultation time, starting from when a patient entered the consultation room and ending when they exited, and non-consultation time during which no patients were present in the consultation room and administrative tasks were recorded. Apart from the length of consultation, the health conditions consulted, reasons for the visit, and any further referrals are noted. As the focus of our manuscript is the FD's and MA's time use, we do not delve into the specific discussions that take place during the consultations. For more detailed insights, please refer to our previously published manuscript [19], see Text in S2 Text.

## Data collection

The data collection for this study took place from 3rd October to 4th November 2022. Data were gathered by eight medical observers, who received three days of training on the study objectives and tools, followed by a pretest and subsequent discussions and training reinforcement. The observers were instructed to shadow the HP within for their entire workday, with minimal interference in the HP's work. The observations were conducted over five consecutive days, from Monday to Friday, by the same observer. Due to logistical constraints, HP were not accompanied by the observers during outreach activities. Instead, only the time spent on outreach visits was recorded.

## Study size

The study size was determined as follows: To obtain a realistic representation of PHC consultations, the study aimed to observe at least 50% of the FDs and 30% of the MAs working in each PHC center. However, the actual number of HP observed was contingent upon the availability of FDs and MAs during the days when observers were present at the PHC centers.

No specific targets were set for the number of consultations, as the volume and duration of these consultations were unknown beforehand (Table 1).

## Data management and analysis

Data were entered using Open Data Kit (ODK) v2022.3.6 and analyzed using IBM SPSS Statistics for Windows, Version 23.0. Data were checked for correctness and plausibility. We then applied simple descriptive statistics to analyze the data. The duration of each activity was captured in minutes and seconds, aggregated into the main categories of activities, and presented as hours and percentages. Furthermore, median values were used to report the length of consultations to exclude outliers and provide more reliable time measurements, see data in S1 Dataset.

# Results

## Work time of family doctors

On average, FDs worked 7.61 hours per day, including lunch and coffee breaks, and in total 38.02 hours per week. According to national regulations, the work time of FDs is 35 hours per week, with a 7-hour workday plus an additional 30-minute break, typically allocated for lunch [20,21]. Moldova has introduced a system of contracting healthcare personnel for more than one full-time equivalent (FTE) with the aim of improving patients' access to healthcare services and addressing shortages of healthcare personnel. In these employment contracts the weekly working hours of HPs increase proportionally when an additional 0.25 or 0.5 FTE is added [20,21]. Observations showed that FDs with a 1.0 FTE contract worked on average slightly more than the hours per week required by their contracts. However, FDs contracted at 1.5 FTEs worked considerably fewer hours on average than stipulated in their employment contracts per week (Table 3). Observations were not carried out on Saturdays, as HP indicated that they would not work on Saturdays during the week of observations. Saturdays are typically considered shorter working days in the PHC centers. Therefore, even if some of the HPs had worked on these Saturdays, it would have had a limited influence on the difference between the observed and required working hours.

The workload of FDs varied across the week. Although the number of daily working hours for FDs remained consistent throughout the work week, FDs completed a significantly greater number of activities from Monday to Wednesday. This indicates that FDs experience a higher workload at the beginning of their work week, which then gradually decreases as the week progresses (Table 4).

## Main activities of family doctors

Nearly half of FDs' work time was dedicated to direct patient care, while a fifth was consumed by administrative tasks, including filling in medical records, preparing documents,

**Table 3. Contract types (full-time equivalents, FTEs) for family doctors: required weekly working hours according to employment contracts, observed mean working hours, and their differences.**

| Contract type (FTEs) | Family doctors | Hours required | Hours observed (mean) | Difference, hours |
|---|---|---|---|---|
| 0.75 | 1 | 26.15 | 28.13 | 1.98 |
| 1.00 | 14 | 35.00 | 36.09 | 1.09 |
| 1.25 | 1 | 43.45 | 42.73 | −0.72 |
| 1.50 | 8 | 52.30 | 42.06 | −10.24 |

**Table 4. Family doctors' average work hours, activities carried out, and their distribution per weekday.**

| Weekday | Work hours/ FD/ day | Activities/ FD/ day | % activities |
|---|---|---|---|
| Monday | 7.86 | 38 | 23.1 |
| Tuesday | 7.97 | 34 | 20.6 |
| Wednesday | 7.89 | 34 | 21.1 |
| Thursday | 7.82 | 30 | 18.5 |
| Friday | 7.79 | 27 | 16.7 |

and scheduling referrals. FDs allocated a tenth of their work time to outreach activities like home visits, though travel took almost twice as long as the home consultations themselves. The majority of personal time was spent on lunch breaks, with the median lunch break lasting 46.63 minutes, exceeding the 30-minute duration specified in the Labor Code of the Republic of Moldova [20]. Less time was spent on other activities such as meetings, medical education, and work-related phone calls. There were unobserved gaps between observations, when the activity was not recorded. Table 5 provides a breakdown of activity volumes and hours.

## Health conditions consulted by family doctors

Consultations observed in this study were divided two health condition groups: 1) non-communicable diseases (NCDs) including; hypertensive diseases, diabetes, ischemic heart diseases, overweight, obesity and 2) non-NCDs (Table 6). Hypertensive diseases were the most frequent and time-consuming health conditions, using nearly half of FDs' consultation time. Hypertensive patients were defined based on the national clinical protocols for PHC approved by Ministry of Health of Moldova. For further details, refer to our previously published paper [19]. Pediatric consultations accounted for the second largest share of the workload. Other health conditions each consumed less than five percent of FDs' work time. This distribution highlights the diverse clinical expertise required

**Table 5. Work time allocation across activity categories by family doctors.**

| Type of activity | n | hours | % of time |
|---|---|---|---|
| Direct patient care | 2,677 | 500.2 | 49.2 |
| Administration | 790 | 207.8 | 20.4 |
| Outreach activities | 100 | 99.6 | 9.8 |
| Personal time | 129 | 84.1 | 8.3 |
| Meeting | 43 | 18.1 | 1.8 |
| Medical education | 6 | 14.6 | 1.4 |
| Work phone calls | 193 | 13.1 | 1.3 |
| Other | 12 | 13.3 | 1.3 |
| Unobserved | | 65.4 | 6.4 |
| **Total** | 3,950 | 1,016.2 | 100.0 |

Note: Each activity includes sub-categories of activities. 1) Direct patient care: NCD consultations, non-NCD consultations. 2) Administration: filling in medical records, statistical reporting/health information system, preparing documents and scheduling referrals, filling in forms, planning and organization of health services, getting laboratory/investigation results, organizing/getting patient cards, making copies, other. 3) Outreach activities: home visits, travel related to home visits. 4) Personal time: lunch break, tea/coffee break, waiting for a patient, personal time, absence with no reason. 5) Meeting: Medical stuff meeting, one to one meeting with a staff member, pharmaceutical representative, district health management team meeting. 6) Medical education: giving training to staff, giving training to patients, participating in a training workshop, medical reading/self-study. 7) Work related phone calls: phone consultation with a patient, other calls.

**Table 6. Health conditions and work time allocation across consultations by family doctors, and the distribution between NCDs and other health conditions.**

| Health conditions consulted | n | Hours | % of time | Median (minutes) | SD |
|---|---|---|---|---|---|
| Hypertensive disease (NCD) | 1,039 | 232.8 | 46.5 | 11.33 | ± 11.21 |
| Pediatric consult | 754 | 117.6 | 23.5 | 7.73 | ± 9.48 |
| Musculoskeletal system | 166 | 25.7 | 5.1 | 6.68 | ± 9.30 |
| Respiratory system | 133 | 20.0 | 4.0 | 7.02 | ± 7.75 |
| Pregnancy consult | 93 | 20.1 | 4.0 | 10.03 | ± 11.81 |
| Digestive system | 94 | 18.7 | 3.7 | 8.95 | ± 14.46 |
| Diabetes (NCD) | 65 | 11.0 | 2.2 | 8.22 | ± 6.83 |
| Ischemic heart diseases (NCD) | 22 | 5.0 | 1.0 | 9.71 | ± 10.92 |
| Sexual and reproductive health | 31 | 5.1 | 1.0 | 6.97 | ± 9.37 |
| Endocrine diseases (excluding diabetes) | 26 | 4.1 | 0.8 | 7.45 | ± 6.84 |
| Visual system | 17 | 2.2 | 0.4 | 5.48 | ± 5.70 |
| Unclear | 20 | 1.7 | 0.3 | 3.44 | ± 5.04 |
| Overweight, obesity (NCD) | 3 | 0.4 | 0.1 | 5.48 | ± 3.29 |
| Other | 214 | 35.8 | 7.2 | 7.30 | ± 12.41 |
| **Total** | **2,677** | **500.2** | **100.0** | **8.85** | **± 10.71** |
| NCD consultations | | | | | |
| Patients with NCD | 1,129 | 249.2 | 49.8 | 10.98 | 11.01 |
| Patients with non-NCD | 1,548 | 251.0 | 50.2 | 7.48 | 10.23 |

of FDs. Table 6 providers an overview of the health conditions and work time allocation across consultations by FDs.

The median duration of FD consultations was 8.85 minutes per consultation. With 49.2% of their work time dedicated to direct patient care, this means that, on average, FDs consulted 25 patients per day. Consultation lengths varied notably; hypertensive disease consultations were the longest, over eleven minutes, reflecting the complexity and time required to follow the PEN protocol [22]. In contrast, conditions such as musculoskeletal or respiratory system issues had shorter durations, closer to seven minutes.

FDs spent their consultation time almost equally between patients with NCDs and those with health conditions other than NCDs. Interestingly, in addition to hypertensive disease, other NCD conditions; diabetes, ischemic heart diseases, overweight, and obesity, accounted together only for less than 3.5% of FDs' consultation time.

## Reasons for visits consulted by family doctors

Patients' reasons for visit reveals key insights into the workload of FDs. The onset of new symptoms was the most frequent and time-consuming reason for visit. This was followed by treatment evaluations and prescriptions of medication. Together these reasons account for the majority of the FDs' workload. Table 7 shows an overview of the reasons for visits and work time allocation by FDs.

## Work time of medical assistants

On average, MAs worked 7.71 hours per day, including lunch and coffee breaks, and in total 37.74 hours per week. Observations showed that MAs with a 1.0 FTE contract worked on average slightly more than the hours per week required by their employment contracts. However, MAs contracted at 1.5 FTEs worked considerably fewer hours on average than stipulated in their employment contracts per week (Table 8).

**Table 7. Reason for visit and work time allocation across consultations by family doctors.**

| Reason for visit | n | hours | % of time | Median (minutes) | SD |
|---|---|---|---|---|---|
| Onset of new symptoms | 1,037 | 220.1 | 44.0 | 9.83 | ±12.63 |
| Treatment evaluation | 755 | 137.5 | 27.5 | 9.17 | ± 8.60 |
| Prescription of medication | 307 | 55.7 | 11.1 | 9.23 | ± 8.55 |
| Specialist referral | 166 | 27.8 | 5.6 | 7.43 | ± 13.03 |
| Referrals to investigations | 195 | 27.5 | 5.5 | 7.03 | ± 6.33 |
| Medical documentation | 176 | 24.0 | 4.8 | 5.23 | ± 9.94 |
| Medical emergency | 6 | 1.5 | 0.3 | 8.78 | ± 17.18 |
| Pregnancy monitoring | 22 | 4.1 | 0.8 | 9.99 | ± 8.13 |
| Unclear | 13 | 2.0 | 0.4 | 4.98 | ± 10.95 |
| **Total** | **2,677** | **500.2** | **100.0** | **8.85** | **± 10.71** |

**Table 8. Contract types (full-time equivalents, FTEs) for medical assistants: required weekly working hours according to employment contracts, observed mean working hours, and their differences.**

| Contract type (FTEs) | Medical assistants | Hours required | Hours observed (mean) | Difference, hours |
|---|---|---|---|---|
| 1 | 17 | 35.0 | 36.05 | 1.05 |
| 1.5 | 7 | 52.3 | 41.83 | −10.47 |

The workload of MAs varied across the week. Although the number of daily working hours for MAs increased slightly towards the end of week, MAs completed a significantly greater number of activities from Monday to Wednesday. This indicates that MAs experience a higher workload at the beginning of their work week, which then gradually decreases as the week progresses (Table 9).

## Main activities of medical assistants

The largest share of MAs' work time was allocated to administration, predominantly patient administration with medical record filling. Surprisingly, direct patient care was only the second largest component of their duties. This was followed by outreach activities, where the majority of time was spent traveling to home consultations. The majority of personal time was spent on lunch breaks, with the median lunch break lasting 48.13 minutes, exceeding the 30-minute duration specified in the Labor Code of the Republic of Moldova [20]. Less time was spent on other activities such as meetings, medical education, and work-related phone calls. Additionally, there were unobserved gaps between the activities when the time was not recorded. Table 10 details the distribution of activity volumes and hours for MAs.

**Table 9. Medical assistants' average work hours, activities carried out, and their distribution per weekday.**

| Weekday | Work hours/ MA/ day | Activities/ MA/ day | % activities |
|---|---|---|---|
| Monday | 7.61 | 51 | 24.4 |
| Tuesday | 7.61 | 46 | 21.7 |
| Wednesday | 7.61 | 43 | 20.4 |
| Thursday | 7.94 | 38 | 18.1 |
| Friday | 8.70 | 32 | 15.4 |

**Table 10. Work time allocation across activity categories by medical assistants.**

| Type of activity | n | Hours | % of time |
|---|---|---|---|
| Direct patient care | 2,239 | 301.9 | 30.9 |
| Administration | 1,770 | 328.4 | 33.6 |
| Outreach activities | 117 | 132.5 | 13.6 |
| Personal time | 200 | 95.3 | 9.8 |
| Meeting | 184 | 28.2 | 2.9 |
| Medical education | 42 | 5.1 | 0.5 |
| Work phone calls | 472 | 16.3 | 1.7 |
| Other | 36 | 14.3 | 1.5 |
| Unobserved | | 55.2 | 5.6 |
| **Total** | **5,060** | **977.0** | **100.0** |

Note: Each activity includes sub-categories of activities. 1) Direct patient care: NCD consultations, non-NCD consultations. 2) Administration: filling in medical records, statistical reporting/health information system, preparing documents and scheduling referrals, filling in forms, planning and organization of health services, getting laboratory/investigation results, organizing/getting patient cards, making copies, other. 3) Outreach activities: home visits, travel related to home visits. 4) Personal time: lunch break, tea/coffee break, waiting for a patient, personal time, absence with no reason. 5) Meeting: Medical stuff meeting, one to one meeting with a staff member, pharmaceutical representative, district health management team meeting. 6) Medical education: giving training to staff, giving training to patients, participating in a training workshop, medical reading/self-study. 7) Work related phone calls: phone consultation with a patient, other calls.

## Health conditions consulted by medical assistants

Similarly, to FDs, MAs allocated a significant portion of their direct patient care time to hypertensive consultations. Pediatric consultations also comprised a large part of their direct patient care work time, followed by pregnancy and musculoskeletal consultations, each requiring a smaller fraction of the MAs' time. Table 11 provides a breakdown of the health conditions and time allocation across consultations by MAs.

**Table 11. Health conditions and work time allocation across consultations by medical assistants, and the distribution between NCDs and other health conditions.**

| Health conditions consulted | n | Hours | % of time | Median (minutes) | SD |
|---|---|---|---|---|---|
| Hypertensive disease (NCD) | 959 | 149.6 | 49.6 | 6.75 | ± 9.92 |
| Pediatric consult | 571 | 66.2 | 21.9 | 4.73 | ± 6.38 |
| Pregnancy consult | 79 | 13.5 | 4.5 | 6.30 | ± 9.63 |
| Musculoskeletal system | 109 | 12.5 | 4.1 | 5.98 | ± 5.24 |
| Digestive system | 91 | 11.6 | 3.8 | 5.40 | ± 6.39 |
| Respiratory system | 105 | 11.5 | 3.8 | 4.28 | ± 6.02 |
| Endocrine diseases (excluding diabetes) | 28 | 3.5 | 1.2 | 4.35 | ± 6.34 |
| Diabetes (NCD) | 29 | 3.3 | 1.1 | 4.77 | ± 6.34 |
| Sexual and reproductive health | 24 | 2.8 | 0.9 | 4.69 | ± 7.67 |
| Visual system | 18 | 2.5 | 0.8 | 7.26 | ± 5.37 |
| Ischemic heart diseases (NCD) | 21 | 2.1 | 0.7 | 4.60 | ± 4.06 |
| Overweight, obesity (NCD) | 11 | 1.2 | 0.4 | 6.33 | ± 4.42 |
| Unclear | 19 | 0.8 | 0.3 | 1.78 | ± 2.26 |
| Other | 175 | 20.8 | 6.9 | 4.13 | ± 9.16 |
| **Total** | **2,239** | **301.9** | **100.0** | **5.65** | **± 8.41** |
| **NCD consultations** | | | | | |
| Patients with NCD | 1,020 | 156.2 | 51.7 | 6.56 | ± 9.73 |
| Patients with non-NCD | 1,219 | 145.7 | 48.3 | 4.82 | ± 7.00 |

The median duration of MA consultations was short, 5.65 minutes. With 30.9% of their work time dedicated to direct patient care, this means that, on average, MAs consulted 25 patients per day. The duration of consultations for hypertensive disease was slightly longer, due to their complexity and the time required to follow the PEN protocol [22].

MAs spent their consultation time almost evenly between patients with NCDs and those presenting with other health conditions (Table 11). Notably, aside from hypertensive diseases, other NCDs; diabetes, ischemic heart diseases, overweight, and obesity, accounted for only a minor share of the MAs' consultation time.

## Reasons for visits consulted by medical assistants

The onset of new symptoms was the most frequent and time-consuming patients' reason for visits to MAs, followed by medication prescriptions. Although MAs are not allowed to prescribe medication, FDs often delegate the administration of this task to MAs. Treatment evaluations also accounted for a significant part of their workload. Table 12 provides an overview of the reasons for visits and work time allocation by MAs.

## Electronic medical records system

The electronic medical records system (SIA AMP) was introduced in Moldova already in 2019 [23]. However, the results showed that only a quarter of FDs used the system for every patient consultation, and nearly half used it solely for reporting (Table 13). The situation was slightly better with MAs; 41.7% used the system for every patient consult, and a quarter for reporting only. This indicated that most PHC centers use an inefficient combination of paper and electronic reporting. Alarmingly, about a third of healthcare workers did not use the electronic system at all.

**Table 12. Reason for visit and work time allocation across consultations by medical assistants.**

| Reason for visit | n | hours | % of time | Median (minutes) | SD |
|---|---|---|---|---|---|
| Onset of new symptoms | 615 | 91.7 | 30.4 | 5.58 | ±10.89 |
| Prescription of medication | 482 | 59.3 | 19.6 | 6.26 | ± 5.46 |
| Treatment evaluation | 300 | 56.6 | 18.7 | 9.18 | ± 9.84 |
| Referrals to investigations | 385 | 42.5 | 14.1 | 4.60 | ± 6.67 |
| Specialist referral | 238 | 31.8 | 10.5 | 5.81 | ± 7.18 |
| Medical documentation | 182 | 15.8 | 5.2 | 3.30 | ± 5.46 |
| Medical emergency | 8 | 1.7 | 0.6 | 11.64 | ± 8.26 |
| Pregnancy monitoring | 15 | 1.2 | 0.4 | 3.55 | ± 3.46 |
| Unclear | 14 | 1.3 | 0.4 | 3.00 | ± 5.45 |
| **Total** | **2,239** | **301.9** | **100** | **5.65** | **± 8.41** |

**Table 13. Use the electronic medical records system (SIA AMP) by family doctors and medical assistants.**

| Use the electronic medical records system* | Family doctors | | Medical assistants | |
|---|---|---|---|---|
| | n | % | n | % |
| For every patient consult | 6 | 25.0 | 10 | 41.7 |
| For reporting only | 11 | 45.8 | 6 | 25.0 |
| Does not use | 5 | 20.8 | 4 | 16.7 |
| The system is not functional in HC | 2 | 8.3 | 4 | 16.7 |

*Sistemul Informațional Automatizat pentru Asistența Medicală Primară (SIA AMP).

## Discussion

This study provides valuable insights into the real-world time-use of healthcare personnel in PHC in Moldova. The results indicate that FDs spent half of their time on direct patient care and a fifth on administration. Whereas MAs spent a third of their work time on administrative tasks and only a third on direct patient care. FDs and MAs spent half of their direct patient care time to NCD consultations and the other half to those presenting with other health conditions. The median consultation durations were short: FDs' consultations were 8.55 minutes and MAs' only 5.65 minutes. Hypertensive diseases and pediatric visits were the most time-consuming consultations. Onset of new symptoms and medication prescriptions were the most frequent reasons for PHC visits. The results indicate that both FDs and MAs contracted for 1.5 FTEs worked approximately 10 hours per week shorter than required by their employment contracts.

A positive result is the substantial portion of FDs' work time dedicated to direct patient care. However, the short duration of consultations combined with a considerable amount of time allocated to administrative tasks suggests that FDs are burdened with a high patient load and administration, potentially compromising the quality of consultations. It's positive that MAs handle a larger share of administrative tasks, potentially lightening the administrative workload for FDs. However, the small proportion of MAs' work time spent on direct patient care, coupled with the notably short duration of these consultations, raises concerns about the quality of care MAs provide. This is surprising, considering that MAs' role is to support by handling routine diagnostic tasks and engaging more extensively with patients, which should allow FDs to concentrate on interpreting examination results and the health conditions of the patients. The findings suggest a weak involvement of MAs in delivering the full range of clinical services at the PHC level in Moldova. Currently, measures to strengthen MA's role in PHC are undertaken by Nicolae Testemitanu State University of Medicine and Pharmacy, Republic of Moldova, such as training and education programs to improve their clinical skills and broaden their spectrum of responsibilities. These measures aim to eventually improve the division of clinical workload between FDs and MAs.

Surprisingly, hypertension and pediatric consultations represented approximately 70% of HPs' time used for direct patient care. Hypertensive consultations accounted for 46.5% for FDs and 49.6% for MAs of consultation time. This suggests the high prevalence of hypertension in the population and the impact of PEN protocols in increasing attention towards hypertension management [24]. Additionally, PEN-trained medical personnel tend to perform more comprehensive assessments, which require more time. On the other hand, other NCDs; diabetes, ischemic heart diseases, overweight and obesity, occupied only minimal proportions of FDs' (2.3%) and MAs' (4.5%) consultation time, despite the high prevalence of these diseases [25], indicating limited attention to these conditions. Pediatric consultations consumed a significant portion of FDs' (23.5%) and MAs' (21.9%) consultation time. This likely results from the mandatory practice introduced during the Covid-19 epidemic, requiring FDs to issue absenteeism certificates for children in kindergarten and school after illnesses lasting three days. Earlier, this practice was optional [26]. Additionally, the study took place in autumn, which may have increased pediatric visits due to the seasonal variation of respiratory infections.

HPs allocated a substantial portion of their work time to administrative tasks, with MAs (36.3%) and FDs (22.8%) engaging in tasks such as filling in medical records, preparing documents, and scheduling referrals. An electronic reporting system (SIA AMP) has been introduced in PHC in Moldova in 2018 [23]. However, the results showed that only 25.0% of FDs and 41.7% of MAs used it for every patient consultation, and approximately one third of the HPs did not use the electronic system at all. This dual reporting, both manual and electronic, is likely to reduce the

efficiency and quality of medical reporting. In April 2024, the National Health Insurance Company, in partnership with the Ministry of Health, launched the ePrescription system for prescribing and dispensing compensated medicines and medical devices [27]. This electronic system aims to eliminate the need for paper forms and thereby improve the efficiency of work time.

To improve patients' access to healthcare services and to cope with shortages of healthcare personnel and, Moldova has introduced a system of granting FDs and MAs employment contracts that exceed 1 FTE. The assumption is that personnel with contracts higher than 1 FTE, will work more and receive higher salaries as compensation for the additional work [20,21]. The results of this study indicate inefficiencies in meeting contracted work as both FDs and MAs with 1.5 FTE contracts appear to work approximately 10 hours (20%) less per week than required, which reduces the overall service delivery within the healthcare centers. This suggests that the extra FTE system does not improve patients' access to the PHC services or resolve personnel shortages in practice. Hence, the system seems merely to serve as a mechanism to provide healthcare personnel with higher salaries.

The finding on FDs allocating their time primarily on direct patient care aligns with previously conducted studies in Sweden and US where PHC physicians spent most of their work-time with patients, respectively 35.9% and 55% [28–30]. While these studies also noted a high administrative burden among physicians (50%), our study found a much lower administrative burden on FDs (22.8%). However, the Swedish study reported similar administration time for nurses compared to our findings (35.7% versus 36.3%), while other studies from Albania, Brazil and Australia reported shorter lower portion of administration tasks by nurses, respectively 21.1%, 22.1% and 28.8% [8,31,32]. Furthermore, the MAs' weak involvement in delivering the full range of clinical services finding is supported by findings in Albania and Bosnia and Herzegovina [8,33]. According to these studies nurses' lesser clinical role was caused by the imposed secretarial role and uncertainty with clinical reasoning and skills. Lastly, publications from LMIC countries; Tanzania, Cameroon and Albania, highlighted the prevalence of unproductive time among HP, ranging from 43% to 73% [7,8,34]. Interestingly, our study revealed minimal unproductive time spent by HP, specifically 3.5% for FDs and 4.3% for MAs.

This study had a few limitations. Firstly, the voluntary non-randomized participation of FDs and MAs introduces a selection bias, as those who chose to participate may perform better. Secondly, the study was subject to the Hawthorne effect [35], wherein HP may have modified their working behavior due to the awareness of being observed, informed in advance about the timing of these observations. This could potentially lead to more diligent consultations and a reduction in personal and unproductive time. To minimize the Hawthrone effect, the HP were observed over several consecutive days to acclimate them to the observers' presence and restore their usual behavior. Thirdly, due to logistical constraints, observers did not follow HP during outreach visits. Therefore, the data obtained from these home visits should be interpreted with caution. Fourth, observations were not conducted on Saturdays, which are shorter working days in PHC centers, potentially influencing the difference between the observed and required working hours. Lastly, the cross-sectional study design and the short duration of the study might not represent the annual utilization of HP. Despite these limitations, this study offers valuable insights into the work-time allocation of FDs and MAs in PHC centers in Moldova.

## Conclusions

This study provided unique insights to the time use of health personnel in PHC centers in Moldova. Family doctors used nearly half of their work time on direct patient care, while medical assistants spent less than one third, with administrative tasks taking up most of medical assistants' time. During their direct patient care time, family doctors and medical assistants

spent nearly equal portions on NCD and non-NCD consultations. Consultation durations were short for both cadres, especially among medical assistants, reflecting the high workload in PHC in Moldova. Additionally, health personnel contracted for 1.5 full-time equivalents worked consistently fewer hours than required by their contracts, indicating this system does not improve patient access to healthcare or resolve personnel shortages.

## Supporting information

**S1 Dataset. All the manuscript tables.**
(CSV)

**S2 Text. Related manuscript.**
(PDF)

## Acknowledgments

The authors wish to thank the Moldovan Government, in particular the Ministry of Health and district authorities for their overall facilitation of the study. The time and commitment of the health workers and patients that agreed to participate is deeply appreciated. We would like to thank Natalia Zarbailov for the input on the study design phase. Special thanks to Cristina Rotaru for coordinating the data collection and for Daniel Bekele for analyzing the data.

## Author contributions

**Conceptualization:** Altiona Muho, Jari Kempers, Alexandra Topa, Helen Prytherch.

**Data curation:** Jari Kempers.

**Formal analysis:** Jari Kempers.

**Funding acquisition:** Helen Prytherch.

**Methodology:** Altiona Muho, Jari Kempers, Alexandra Topa.

**Project administration:** Jari Kempers, Helen Prytherch.

**Validation:** Altiona Muho, Jari Kempers, Alexandra Topa, Ala Curteanu, Ghenadie Curocichin, Helen Prytherch.

**Visualization:** Altiona Muho, Jari Kempers.

**Writing – original draft:** Altiona Muho, Jari Kempers, Alexandra Topa.

**Writing – review & editing:** Altiona Muho, Jari Kempers, Alexandra Topa, Ala Curteanu, Ghenadie Curocichin, Helen Prytherch.

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
