## [Decision Letter · Decision Letter 0]

30 Dec 2024

PONE-D-24-21504Time-motion study in Primary Health Care in Moldova: How do family doctors and medical assistants spend their work time?PLOS ONE

Dear Dr. Muho,

Thank you for submitting your manuscript to PLOS ONE. After careful consideration, we feel that it has merit but does not fully meet PLOS ONE’s publication criteria as it currently stands. Therefore, we invite you to submit a revised version of the manuscript that addresses the points raised during the review process.

A very interesting study to see the workload of family doctors in PHC as the backbone of healthcare in every country. From the study, we can see that as a doctor, you are not only provide services to patients but are also inseparable from the demands of administrative workload and community service outside the building, because that is the essence of a doctor's job especially in PHC. This study has given us information regarding the amount of time needed in one working hour cycle to do the three things above. However, it would be better if you could add information regarding the following matters:

1. Why do we need to know the work-time allocation of FD and MA? For what purposes is this data needed? Is it part of some development roadmap?

2. Number of patients per doctor per day

3. Number of working hours per day and per week

4. Office hours or shift work

5. Situation: outpatient clinic only, or includes emergency room and inpatient care?

6. The reason why it is taken 5 days in a row, does it include weekdays and/or weekends

7. What do doctors and medical assistants do during the 8 minute and 5-minute consultations? � Can you explain how long it takes for history taking, physical examination, and education in one consultation?

8. Before meeting the doctor, is there a nurse station that has taken previous clinical data? Was it part of the whole consultation process?

9. Are there other health personnel who work as part of the patient examination series at the PHC?

10. Why are hypertensive and pediatric visits the most frequent and time consuming? What is discussed during the consultation meeting? Does it only depend on the doctor-patient interaction? Or are there other collaborative services?

11. What are the specific implications/impacts on services inside and outside the building from Health personnel contracted for 1.5 full-time equivalents worked over 10 hours per week shorter than required by their working contracts?

12. Table 1. What time-period data?

13. Observation mechanism? Timer used, if any? Who is involved in data collection?

14. What are the differences in the background, qualifications and job descriptions of FD and MA? Because experience in medicine are not much different (11 years vs 10 years)

15. Calculation of sample size: to ensure that 24 FD and 24 adequately represent the population

We look forward to receiving your revised manuscript.

Kind regards,

Retno Asti Werdhani, M.Epid, Ph.D

Academic Editor

PLOS ONE

2. Please provide additional details regarding participant consent. In the ethics statement in the Methods and online submission information, please ensure that you have specified what type of consent you obtained (for instance, written or verbal, and if verbal, how it was documented and witnessed). If your study included minors, state whether you obtained consent from parents or guardians. If the need for consent was waived by the ethics committee, please include this information.If you are reporting a retrospective study of medical records or archived samples, please ensure that you have discussed whether all data were fully anonymized before you accessed them and/or whether the IRB or ethics committee waived the requirement for informed consent. If patients provided informed written consent to have data from their medical records used in research, please include this information

Reviewers' comments:

Reviewer's Responses to Questions

**Comments to the Author**

1. Is the manuscript technically sound, and do the data support the conclusions?

Reviewer #1: Yes

Reviewer #2: Yes

2. Has the statistical analysis been performed appropriately and rigorously? 

Reviewer #1: Yes

Reviewer #2: Yes

3. Have the authors made all data underlying the findings in their manuscript fully available?

Reviewer #1: No

Reviewer #2: No

4. Is the manuscript presented in an intelligible fashion and written in standard English?

Reviewer #1: Yes

Reviewer #2: Yes

5. Review Comments to the Author

Reviewer #1: Overall, the study was written in good English, easy to digest and important for health care manager. The topic might be important only for that particular area, being investigated. However, the idea and the way to observe are actually good for health managers to replicate in their own areas. Therefore, authors should reveal a little bit in the methodology related with operational definition, i.e. how to measure duration, which one is the starting point to observe and to conclude as end consultation time. How do authors define hypertension patients? perhaps the one who need consultation with doctors were the newly diagnosed, or patients with complications, whereas MAs only doing the follow-up, blood pressure monitoring, etc. How did the interprofessional collaboration between MAs and FDs? Whether they work simultaneously, first MA to gather information continues with FDs? How is the working hours? Is there any specific regulation for working hours, i.e. 8 hours/day including rest? How about working hours for doctors? those things that I thought important to describe further the situation of the GPs.

Reviewer #2: This article tries to describe the daily activity of Family Doctor and Medical assistant, in the Primary Healthcare setting. As commonly known, work in PHC setting is rigorous, and a lot of time and energy should be allocated by those working in the field. Thus in this context, the article confirms the presupposition, and no really new information is given.

However in my opinion, the work of family doctor is always fascinating to look at, since they serve the general population with a wide variety of complaints and illness, and often their work is complicated since besides clinical aspects, they are also responsible for administrative aspects. So in this context it is interesting to see how they manage time, and how much time is allocated for several activities. From the article we could see how the PHC system in Moldova works, and compare with our own to see which has a better impact on the society.

I would suggest several points for the author as to make the explanation clearer for readers from different health systems:

1. It would be more interesting if in the introduction the author describes from community perspectives and from family doctor perspectives, what is the impact of the condition found in Moldova, e.g. patient satisfaction, doctor satisfaction, doctor turnover, patient complaints etc.

2. Limit number of tables. Please follow author instruction, and select tables that provide important information.

3. Since not all readers know the educational background of Medical Assistance and their job description, it would be good to briefly describe it.

4. The format should be modified so that line number does not appear. It really disturbs and needs more efforts to read.

Overall, this article is interesting and should be published.

6. PLOS authors have the option to publish the peer review history of their article (what does this mean? ). If published, this will include your full peer review and any attached files.

**Do you want your identity to be public for this peer review?** For information about this choice, including consent withdrawal, please see our Privacy Policy .

Reviewer #1: **Yes: ** Trevino A. Pakasi

Reviewer #2: **Yes: ** Herqutanto

---

## [Author Response · Author response to Decision Letter 1]

24 Jan 2025

The authors would like to sincerely thank the reviewers and the editor for their time and insightful comments and suggestions, which enable us to improve the presentation of our study and the quality of our manuscript. Please find below our point-by-point responses to the concerns raised by the reviewer and the editor and how we addressed these concerns.

1. Why do we need to know the work-time allocation of FD and MA? For what purposes is this data needed? Is it part of some development roadmap? We thank the reviewers for bringing this point to our attention. We updated the introduction section to answer this requirement with the following statement (rows 101-107): “This time motion study is essential for understanding how PHC personnel utilizes the time, highlighting inefficiencies, workload distributions and the time required for various clinical and administrative tasks. These findings can inform targeted interventions by policymakers and healthcare managers to optimize time use, reduce administrative burdens and enhance quality of care. Moreover this study contributes to the evidence base needed to address workforce shortages and improve overall PHC efficiency in Moldova”.

2. Number of patients per doctor per day? Thank you for raising this value point. Calculations of the average number of patients consulted by FDs and MAs have been added to the manuscript in rows 278-280 and 346-348 for clarity and reference.

Both FDs and MAs consulted 25 patients per day.

3. Number of working hours per day and per week? We thank the reviewer for this comment. The manuscript provides detailed information on the working hours of FDs and MAs. On average, FDs worked 7.61 hours per day and 38.02 hours per week (outlined in rows 218-2019), while MAs worked 7.71 hours per day and 37.74 hours per week (outlined in rows 302-303).

4. Office hours or shift work? We would like to thank the reviewer for bringing up this essential point. We have now added the following sentences in rows 147-149:

In the observed PHCs in Moldova, FDs and MAs primarily operated within fixed office hours rather than shift work. Their schedules typically followed a standard workday structure, with occasional variations for specific tasks or patient demands.

5. Situation: outpatient clinic only, or includes emergency room and inpatient care? Thank you for bringing this aspect to our attention. The study was conducted exclusively in outpatient PHCs and did not include emergency room or inpatient care services. This is now specified in the text on row 110.

6. The reason why it is taken 5 days in a row, does it include weekdays and/or weekends? We would like to thank the reviewer for this valid question. Observations did not include weekends, as the observed medical personnel did not work on Saturdays. This is reflected in the rows 228-230 in the manuscript. Further, the lack of observation on weekends has been listed as a limitation of the study, see lines 467-469. Further, the observations were conducted over 5 consecutive days to minimize the Hawthrone effect. The detailed explanation is outlined in rows 463-465.

7. What do doctors and medical assistants do during the 8 minute and 5-minute consultations? � Can you explain how long it takes for history taking, physical examination, and education in one consultation? We recognize the importance of the reviewers’ raised point. The observations covered the time from when the patient entered the consultation room until the patient exited. However, due to the wide variety of health conditions managed in PHC, it was not feasible to conduct more detailed recordings of specific activities during each consultation. The only type of data collected during the consultation time was the type of health conditions consulted, the reasons for the visit and whether there was any further referrals. This information is outlined in the rows 189-192.

8. Before meeting the doctor, is there a nurse station that has taken previous clinical data? Was it part of the whole consultation process? We acknowledge the comment. In the observed PHCs, MAs supported FDs by carrying out prophylactic, diagnostic and curative assessments, managing patient records and preparing the workplace and instruments. MAs in Moldova are comparable to PHC nurses in other health systems. This information is outlined in lines 136-142.

9. Are there other health personnel who work as part of the patient examination series at the PHC? Thank you for your questions. While more health personnel is involved in the patient examination series at the PHC, our study specifically focused on FDs and MAs and we did not collect data on other. This is now outlined in the manuscript in rows 129-131.

10. Why are hypertensive and pediatric visits the most frequent and time consuming? What is discussed during the consultation meeting? Does it only depend on the doctor-patient interaction? Or are there other collaborative services? Thank you for your questions. Hypertensive visits were the most frequent and time-consuming due to the high prevalence of hypertension in the population. This reflects the impact of PEN protocols in increasing attention to hypertension management. Additionally, PEN-trained medical personnel tend to perform more comprehensive assessments, which require more time. Pediatric visits were also influenced by the mandatory practice of issuing absenteeism certificates for children after illnesses. This information is outlined in rows 407-419 of the manuscript. Regarding the rest of the questions, as the focus of our manuscript is the FD’s and MA’s time use, we do not delve into the specific discussions that take place during the consultations. For more detailed insights, we provide the reference of our previously published manuscript. This information is outlined in rows 189-192. We hope this explanation meets the reviewer’s requests.

11. What are the specific implications/impacts on services inside and outside the building from Health personnel contracted for 1.5 full-time equivalents worked over 10 hours per week shorter than required by their working contracts? We thank the reviewers for this valid point. The medical personnel with 1.5 FTE contracts worked 10 hours per week less than their contracted hours. There were no observations to suggest that this discrepancy proportionally influenced the outreach work performed by them. However, this finding indicates inefficiencies in meeting contracted work hours, which reduces the overall service delivery within the healthcare centers, as outlined in rows 433-442.

12. Table 1. What time-period data? Thank you for raising this question. Table 1 presents data from the observation period, 3rd of October until 4th of November 2022. A note is now added under Table 1, row 116.

13. Observation mechanism? Timer used, if any? Who is involved in data collection? We acknowledge the reviewers’ comment.

The study utilised purpose-built electronic forms developed and tested specifically for this study. These forms recorded time stamps for the start and end of each activity. The forms were filled out solely by the observers using tablets to ensure accuracy and consistency in data collection, as outlined in rows 168-176. The observers were medical personnel trained specifically for the time-motion study and responsible for conducting the observations, as described in rows 194-197.

14. What are the differences in the background, qualifications and job descriptions of FD and MA? Because experience in medicine are not much different (11 years vs 10 years)? We acknowledge reviewer’s observation and thus added a brief description of the educational background and job descriptions of the health personnel in rows 134-142: FDs are trained physicians with medical degrees and specialized training in family medicine. MAs typically complete vocational or post-secondary education programs focused on primary healthcare. Their roles and responsibilities differ, as is common in healthcare systems where doctors and nurses perform complementary functions. The study did not investigate further into the length of experience of the medical personnel beyond the reported averages.

15. Calculation of sample size: to ensure that 24 FD and 24 adequately represent the population? We thank the reviewer for this important observation. The actual number of HP observed was contingent upon the availability of FDs and MAs during the days when observers were present at the PHC centres. Therefore, no sample size calculations were performed. No specific targets were set for the number of consultations, as the volume and duration of these consultations were unknown beforehand (please refer to rows 203-208).

16. Provide more details on the operational definitions, such as how the consultation duration is measured (e.g., starting and ending points for observation)? We thank the reviewer for pointing this out. The recording of consultation time began when the patient entered the consultation room and concluded when the patient exited the room. The definition is now added in row 185-188.

17. Define how hypertension patients are classified. Are they newly diagnosed, patients with complications, or those requiring follow-ups (e.g., blood pressure monitoring by MAs)? We thank the reviewer for the question. The definition of hypertensive patients follows the national clinical protocols for PHC approved by MoH of Moldova. This addition is outlined now in lines 272-274.

18. Describe the collaboration between MAs and FDs. Do they work simultaneously, or does one (e.g., MA) gather information before the other (e.g., FD) continues the consultation? We acknowledge the importance of this point and added the following explanation in the manuscript in rows 139-142: “MAs and FDs typically collaborate in sequential workflow. MAs handle the above-mentioned preparatory tasks with patients prior to FD’s consultation, allowing FDs to focus on diagnosis, treatment decisions and complex case management during the interactions with patients”.

19. Clarify the regulations on working hours, including whether the 8-hour workday includes rest periods. Thank you for this comment. The following supporting statement can be found in the rows 220-221 According to national regulations, the work time of FDs is 35 hours per week, with a 7-hour workday plus an additional 30-minute break, typically allocated for lunch.

20. Provide specific details on the working hours of doctors and whether any specific guidelines apply. We acknowledge the comment. Family doctors in Moldova are contracted for 35 hours per week for a full-time equivalent (1.0 FTE), with proportional increases for additional FTE contracts (e.g., 1.5 FTE contracts require 52.5 hours per week). The detailed information can be found in rows 218-225 and on Table 3.

21. Enhance the introduction by discussing the impact of PHC conditions in Moldova from both community and family doctor perspectives. We have revised the introduction to include more information on how PHC conditions in Moldova affect both the community and family doctors, now outlined in rows 55-59 and 82-94.

This includes a mention of community reliance on PHC services, as well as the strain on FDs caused by high patient loads. Thank you for the advice. We hope that these few additions contextualize the study more comprehensively.

22. Reduce the number of tables to align with author guidelines. Focus on presenting tables that provide the most critical information. Thank you for this suggestion. The time-motion study generated a large and rich dataset, which we have already significantly narrowed down to tables that represent the essential points of time-use. We believe that further reducing the number of tables would compromise the usefulness of the paper for other studies and publications. In addition, smaller tables, such as Table 3 and Table 8, are important for presenting key findings on work time under different contract types.

23. Briefly describe the educational background and job description of Medical Assistants to help readers unfamiliar with their role understand the study. We thank the reviewer for raising this important point. We have added a brief description of the educational background and job description of MAs to provide clarity for readers in the rows 135-139: In Moldova, MAs typically complete vocational or post-secondary education programs focused on PHC. MAs role is to support FDs by carrying out prophylactic, diagnostic and curative assessments, managing patient records and preparing the workplace and instruments.

24. Modify the format to remove line numbers, as they distract readers and make the article harder to read. We appreciate the reviewers comment. The line numbers are included only in the draft manuscript to assist with commenting and reviewing. These will be removed by the editorial office in the final version.

---

## [Editor Report · Decision Letter 1]

5 Mar 2025

Time-motion study in Primary Health Care in Moldova: How do family doctors and medical assistants spend their work time?

PONE-D-24-21504R1

Dear Dr. Muho,

We’re pleased to inform you that your manuscript has been judged scientifically suitable for publication and will be formally accepted for publication once it meets all outstanding technical requirements.

Kind regards,

Retno Asti Werdhani, M.Epid, Ph.D

Academic Editor

PLOS ONE
---

## [Editor Report · Acceptance letter]

PONE-D-24-21504R1

PLOS ONE

Dear Dr. Muho,

I'm pleased to inform you that your manuscript has been deemed suitable for publication in PLOS ONE. Congratulations! Your manuscript is now being handed over to our production team.

Kind regards,

on behalf of

Dr. Retno Asti Werdhani

Academic Editor

PLOS ONE